



# Morphology of an arid landscape utilising synthetic-aperture radar (SAR) and differential interferometric SAR (DInSAR), southern Riyadh, Central Arabia

Mohamed Daoudi[1], Kamel Hachemi[2], Abdullah O. Bamousa[3]

[1] Department of Geography and GIS, Faculty of Arts and Humanities, King Abdulaziz University, Jeddah, Saudi Arabia.

[2] Physical Geography Laboratory (LGP), UMR 8591, CNRS, University Paris-1 and University Paris-Est, 1 place Aristide Briand, 92195 Meudon Cedex, France.

[3] Department of Geology, Faculty of Sciences, Taibah University, Al-Madinah Al-Munawarah, Saudi Arabia.

*Correspondence to*: Mohamed Daoudi (mdaoudi@kau.edu.sa)

**Abstract**. This study on the southern Riyadh area examines the ERS-1/2 and ENVISAT satellites data' ability of detecting Early Quaternary-Holocene trans-tensional Central Arabian graben system morphology. It also, test the hypothesis of potentially geohazardous arid region for the consequent dissolution-induced collapses and karstifications and possible recent faults reactivation. Eight Single Look Complex (SLC) amplitude images are calibrated, filtered, georeferenced, orthorectified, and filtered at a resolution of 20 metres, and compared with one another by producing 17 diachronic images of the pairs at different 15 intervals (1996, 2003-2005, 2008). The diachronic SAR intensity imageries suggest a downthrown displacement reaching 600 m and eastward tilting at the bottoms of the grabens. Also, the structurally-controlled valleys are developing an eastward-running drainage system towards the oasis of Al-Kharj and capturing an older hydrologic system. Moreover, a 12-year period (1996-2008) of the SAR data was obtained to examine the average annual rate of southern Riyadh's anthropological sprawl, which is estimated at approximately 390 metres/year over the 12 years and constrained by geomorphological features towards the 20 deformed area. DInSAR imageries show the primary results obtained from the 26 May 2004 and 31 Jan. 2005 pair of images, merged with 30 m resolution DEM-SRTM data for the arid region south of Riyadh to eliminate the influence of topography. DInSAR is applied in this study for its ability of detecting small displacements at the centimetre scale (1/2 wavelength). Although the DInSAR's coherence and phase imageries suggest a fairly stable region since the last tectonic and subsequent geomorphic events, erosional and artificial changes are observed, bounded within the valleys and depressions, primarily due to aeolian and 25 fluvial processes and agriculture. It is highly recommended to preserve the area for sustainability and economy.

Keywords: Landscape morphology, SAR, DInSAR, Wadi Awsat, Wadi Nisah, Riyadh City.





## 1 Introduction

Desert, semi-arid, and degraded land areas are considered arid regions, despite the presence of irrigated vegetation and
oasis areas (Gamo et al., 2013). The region of this study has similar conditions, in that it is occupied by an oasis, known as the
Al-Kharj oasis, and wadi of Nisah-Sahaba grabens, surrounded and influenced by the Ad-Dahna desert (Weigermars, 1998). The
abundant groundwater and seasonal hydrological water enhanced the feasibility of circular irrigated farms, accommodated by
the depressions and grabens, to the west of the Al-Kharj oasis (Fig. 1). The region has a dry tropical climate, characterised by
high temperatures in summer, cold temperatures in winter, rainfall scarcity and irregularities, high evaporation rates that exceed
precipitation rates, and low relative humidity. The morphology of the landscape is occupied by the Tuwaiq Mountain plateaus,
south of Riyadh (Central Arabia), steeply cut, incised, and stepped over the Wadi Nisah and Wadi Sahaba grabens. Therefore, it
is a complex mountainous landscape, influenced by omnipresent aeolian (wind-driven) erosional processes coming from the Ad-
Dahna desert. Minor fluvial (water-driven) erosional process is also present, during abrupt flash floods, which supply
groundwater resources, transported by the Wadi Nisah-Sahaba grabens into the Al-Kharj oasis. The degree of material removal
from the base of the slopes by fluvial and aeolian processes is strongly influences the landscape weathering at different temporal
and spatial scales (Migon, 2010).

The landscape morphology of this arid region has evolved through a long period of geological process,
geomorphological evolution, and human processes, during both paleo and recent geologic times. Central Arabia, including the
southern Riyadh region is currently under arid land conditions where aeolian processes are prevailing, even though it was
subjected to a complex morphological landscape evolution. The region underwent significant superposed tectonic and
geomorphic events, starting with the deformation of the interior homocline by central arch development and the Alpine-
Himalayan orogeny during the late Tertiary. Then, it underwent a late Central Arabian graben system and associated left lateral
strike-slip faults during the Quaternary (Bamousa et al., 2017). The Nisah-Sahba trans-tensional fault zone is a large-scale feature
that represents the southern boundary of the East Arabian Block (EAB), which is located in the east of Arabia (Fig. 2). The EAB
was developed during the late Alpine-Himalayan orogeny after the opening of the Red Sea and separation of Arabian and African
plates during the Oligocene, leading to the Arabia/Eurasia collision. Since the middle Pleistocene-Holocene period, rainfall and
groundwater level are higher during pluvial phases, developing interstratal and Dinaric karstification of the carbonate- and
evaporate-bearing rocks that formed Cenozoic depressions, caves, and sinkholes (Bamousa et al., 2014; Bamousa 2018). From
the start of the Holocene to the present, the region has been mainly under aeolian and minor fluvial processes. This long history
of deformation and geomorphic development produced attractive arid landform exposures for geotourism, such as the Tuwaiq
escarpment, caves and sink holes, and sand dune fields. Anthropogenic activity and urban sprawl are increasingly spreading in
all directions from Riyadh (Altuwaijri et al., 2018) and might reach the affected region in the coming years. Therefore, with these
underlying ground conditions that might trigger hazards, urban growth trends must be established in a balanced and considered
manner, creating building plans that guide the urban fabric and determine controls for land use and development of agricultural
land between Riyadh and Al-Kharj cities.



Given the abovementioned background of the study area, multi-temporal monitoring over a short period is essential, and analyses can be made via radar images, which represent an efficient and rapid means of studying these arid landscape phenomena. Radar images have the advantage of being taken in all weather conditions, day, and night, and regardless of cloud cover, unlike platforms that operate within the visible range. The use of satellite radar imagery allows for regular and precise

spatial and temporal monitoring of surfaces. The response of the radar signal is particularly sensitive to surface topography, roughness, and moisture, making it possible to determine any natural or artificial changes that occur between two image acquisitions. Synthetic-aperture radar (SAR) is a form of radar that is used to create two- or three-dimensional landscape images. It provides large image coverage (100 km × 100 km) for both generalised and global studies of regions. SAR sensors measure the two components of the backscattered signal, amplitude and phase, where phase (waves) depends on geometrical

characteristics, such as distance information between the radar and the target. Phase corresponds to the temporal distance between the radar sensor and the target. In this regard, it is used in estimating displacement in interferometric applications; this allows for measuring vertical and horizontal displacements in the order of 1/2 wavelength. The amplitude of an image pixel represents the backscattering capability of the terrain of the corresponding ground pixel to send the incident energy back to the antenna (Zhou et al., 2009). It is directly usable and comparable to the optical image, and is also directly related to the surface conditions, thus,

some surfaces are represented by a high radiometric value, such as water surfaces, whereas dry surfaces appear with low radiometric values. SAR images have wide application in remote sensing and geological and environmental studies of arid and semi-arid lands (Troufleau et al., 1994; Singhroy and Saint-Jean, 1999; Wade et al., 2001; Hachemi et al., 2009; 2010; Hachemi and Thomas, 2013; Hachemi et al., 2014a, b; 2015; 2020). Differential interferometric synthetic-aperture radar (DInSAR) uses two or more SAR satellite images to generate maps of surface deformation or digital elevation, using differences in the phase of

the waves returning to the satellite (e.g. Burgmann et al., 2000). This makes it possible to measure vertical ground displacements up to 28 mm in the case of the C-band. DInSAR can yield significant results between two different dates in the same area; one can measure the extensive, precise, and dense ground changes that occur between these two periods.

Central Arabia, south of Riyadh, between latitudes 24°-25° N and longitudes 46°-47° E, has very little existing work on the use of radar images and their applications in arid landscapes. In this area, shuttle radar images have been used to detect several

geological features in the Arabian Peninsula (Al-Hinai et al., 1997), aeolian sand covers (Dabbagh et al., 1997), and faults and landslides under the Quaternary sands of the Ad-Dahna Desert (Weijermars, 1998). In the field of cartography, Alsalman (2010) showed the usefulness of the interpretation of SAR imagery from the Radarsat-l satellite in mapping applications in Riyadh. Another regional study was conducted by Abdelkareem et al. (2020a) on the integration of multispectral and radar images for geological, geomorphic, and structural study in the Al Qunfudhah region. An additional study by Abdelkareem et al. (2020b)

employed optical and radar images for the identification and monitoring of active/inactive landforms in the driest desert of Saudi Arabia. This study tests the hypothesis of the potential geohazards of an arid region that underwent active trans-tensional tectonism followed by karstification during the Quaternary Era, and has undergone recent renovation and expansion due to population growth. This study tests the ability of recent techniques and methodology to create short-period monitoring and analyses via available SAR and DInSAR images. It utilises these data to summarise the natural and artificial evolution and





changes that had occurred in different areas such as those during a period of almost 12 years. The radar data are provided with colour composition (diachronic) images that represent a 12-year period for monitoring potential hazardous features witnessing several constructions that have multiplied on the periphery of and outside the city of Riyadh. The ultimate goal of this work is to alert the community and preserve the area over the south of Riyadh and east of Al-Kharj, in Central Saudi Arabia, for future water and food security, for economic aspects, and as a national geopark for scientific research and tourism.


**2 Data and methods**

To fulfil the abovementioned objectives, the available radar data are acquired; two radar wave components of ERS-1/2 and ENVISAT images are found between 1996 and 2008 (Table 1). Eight amplitude images are produced, multi-dated, calibrated,
filtered, georeferenced, orthorectified, and filtered at a resolution of 20 metres. These images are compared with one another by producing 17 diachronic images of the pairs at different intervals (1996, 2003, 2004, 2005, 2008). The eight Single Look Complex (SLC) images of SAR satellites ERS-1/2 and ENVISAT were processed by pulse compression in the radial direction in distance and by SAR aperture synthesis in the azimuthal direction. In this type of product, each pixel is represented by a complex value (real and imaginary part) from which the amplitude and phase of the signal are extracted. For this study, eight SLC images from
the ERS-1/2 and ENVISAT satellites' images have 4,904 columns and 29,715/29,708 rows for ERS-1/2 and 5,164 columns and 26,172, 27,314, 27,325, 27,315, 27,325, or 27,326 rows for ENVISAT. Radiometric calibration involves bringing the produced image (amplitude, intensity) to a real ground reflectance or radiance. Geo-referencing makes it possible to have the images produced in the same cartographic projection system (UTM zone 38-N, WGS84). This step was carried out automatically by assigning the geographical characteristics of each image. Orthorectification involves correcting the amplitude or intensity image
to the shape of the Digital Elevation Models (DEM) from the Shuttle Radar Topography Mission (SRTM) at 30 m resolution. Filtering is the removal of the speckle effect to increase the readability of an image. The data experienced radiometric calibration, processed by pulse compression in the radial, range, and azimuthal directions. These images used in this study cover an area of 100 km × 100 km with a resolution of 4 m in azimuth and 20 m in distance. In this type of product, each pixel is represented by a complex value (real and imaginary parts) from which the signal amplitude and phase are extracted. The acquisition is
descending (day). The scene is illuminated to the right in the lateral view with a 23° angle of incidence in the C-band at a wavelength of 5.65 cm and with vertical polarisation (V/V).

The region south of Riyadh using SAR images underwent the application of the derivative of the differential interferometry (DInSAR) technique, the coherence image, which distinguishes stable areas that have retained the phase from non-stable areas that disturb the phase. Figure 3 shows the primary results obtained from the pair of images obtained on 26 May
2004 and 31 Jan. 2005. DEM-SRTM images are used for the arid region south of Riyadh with a resolution of 30 m to eliminate the influence of topography. The choice of the two dates used in the coherence image is based on feasibility conditions; data are considered by temporal baseline interval (Btemp) from one day to 12 years, adopting a spatial baseline of < 1,000 m. This study used the GAMA software, under the Linux and Windows (Cygwin) operating systems, for the realisation of the differential interferograms. All the pairs are formed from these data, and the calculated geometric characteristics of the images for each pair





are processed (Table 2). The DInSAR was performed using the Centre national d'études spatiales (CNES) method, using two passes and an external DEM. The carried out treatments of differential interferograms show fringes corresponding to the phase difference. Coherence images demonstrate the reliability of the interferograms producing:

$$(0 = \text{minimum coherence}; 1 = \text{maximum coherence}) \tag{1}$$

Coregistered amplitude images correspond to the realised differential interferograms and unwinding of the realised phase difference. These images are compared with one another by making colour composites at different intervals ranging from one day (24 hours) to 11 years and seven months (1996-2008); the processing steps perform and extract eight calibrated, georeferenced, orthorectified, and filtered amplitude images of the study area in the southern region of Riyadh (Fig. 4). 17 colour composition pairs are produced for different satellite intervals (ERS and ENVISAT). These intensity (colour composition) image pairs of the study area are shown in Fig. 5. The colour of each pixel of the resulting image was processed by the equation:

$$C(p) = a(p) \, [GB] + b(p) = [R] \tag{2}$$

in which $a(p)$ and $b(p)$ represent values depending on the signal strength of the pixel for the oldest and most recent component respectively; thus, pixel appears in Red [R]. Table 3 summarises the results of colour composite enhancements, while Table 4 shows the legend of the low and high signal and the reflected colour. For a pixel appearing in black and in white, signal strength is equal to:

$$a(p) = 0, \text{ and } b(p) = 1; \text{ hence } C(p) = [R] \tag{3}$$

## 3 Results

As a result of using amplitude, intensity, coherence, and interferometric phase imageries, the following landscapes were distinguished: mountainous areas, depressions and grabens, sand dune areas confirmed by the Landsat imagery (Fig. 6), and areas of human activity (urbanisation, roads, and agricultural fields). The radar data have given rise to several different geomorphological phenomena, such as river captivity and the fragmentation of the surface topography of the plateaus by the grabens and depressions.

### 3.1 Morphology of the landscape

The Tuwaiq Mountain series (TM) form an overall plateau, cut through by different wadis (Fig. 1), such as Wadi Hanifah (WH), which extends from northwest to southeast, and Wadi Al Awsat (WA) and Wadi Nisah (WN), which extend from west to east and empty into a regional depression known as the Al-Kharj Depression (AD). The Wadi Hanifah and Wadi as-Silay (WS) are linear valleys running north to south, whereas Wadi Nisah and Wadi Hanifah run west to east and are originally grabens formed by tectonic events (Bamousa et al., 2017; Bamousa, 2018). These wadis cross-cut steep beds between hard and brittle Mesozoic and Cenozoic clastic, carbonate, and evaporitic rocks (Fig. 2).



### 3.2 Amplitude and intensity images

The amplitude illustrates the specificities of the different landscapes of this region, such as the grabens and the Tuwaiq mountain range. It also shows the overlying urban extension of the city of Riyadh towards the deformed areas (Fig. 7). This
extension is characterised by four directions from the southeast to the southwest, estimated at different rates, related to the encountered complexity of the terrain (Table 5). In the southeast direction, the rate of extension was estimated at approximately 240 m m/year; towards the south, this was derived as approximately 460 m/year; towards the southwest, this rate was approximately 770 m/year, and towards the west-southwest, it was approximately 90 m/year. Thus, the total rate of extension is 1560 m/year, and the average of extension towards the deformed area is 390 m/year during the 12-year period. N-S profiles 1, 2,
and 3, subtracted from the intensity image, show the variation of the topography from north to south (Fig. 8). It shows the effect of the early Quaternary graben system and the consequent late Pleistocene karstifications that deformed the Tuwaiq Mountain series in the study area. The maximum height of elevation is approximately 1,000 m in the west, and the lowest altitude is approximately 400 m in the east, indicating a maximum 600 m of downthrown displacement and depression. Profiles 4 and 5, plotted E-W along the Awsat and Nisah valleys reveal eastward tilting and consequent capturing of the hydrologic system by the
last tectonic event in the Central Arabian graben system (Fig. 8). Therefore, a new hydrologic system has developed, in which Awsat is pouring into the Nisah that in turn pours into the Sahba valleys west of Al-Kharj City (Fig. 9).

### 3.3 Differential interferometry

In the 280-day period noted above, the southern region of Riyadh showed good coherence (yellow colour) in the high relief area and poor coherence (blue-violet colour) in the low relief and agricultural areas (Fig. 9). Coherence represents the pixel stability, which distinguishes high phase values (high coherence) from those that disturb the phase (low coherence), such as vegetation and water, among others (Fig. 9). The analysis of the landscape south of Riyadh indicates stable terrain with no landslides, subsidence, or flash floods suggested during the period 1996-2008. The consistency image made it possible to
distinguish between two parts of the study area. One part retained the signal phase, showing high coherence translated by the Tuwaiq mountain range; in another part, the signal phase was disturbed in e.g. the graben and valleys, indicating low coherence due to unconsolidated sediments, water, and agriculture. The produced coherence image shows the reliability of the differential interferogram performed (Fig. 9). The realised differential interferogram phase imagery does not indicate displacement fringes (Fig. 10). The course of this phase difference indicates the atmospheric phase, which correlates with the topography of the region
southwest of Riyadh (Fig. 10). Altitude values ($x$) are not provided with a numbered scale for unknown coefficient ($k$) value in the following equation:

$$x = 2k\pi. \tag{4}$$

If ($k$) is equal to (1), the lowest altitude value is equal to ($2\pi$). Therefore, the interval of the provided range is equal to 50 m/colour, ranging from 400 m to 1,000 m according to the topographic map (Fig. 10).



### 3.4 Diachronic image interpretation

The use of the images dated 29 Sept. 2003, 12 Jan. 2004, and 26 Apr. 2004, and their colour compositions, aimed to enable observation of changes due to the autumn, winter, and spring seasons, which corresponded to the months of acquisition of these three radar images. These images indicate that the observed changes were mainly due to weather conditions. The Jan data indicate a high signal strength, which can be translated into high humidity due to precipitation in the winter of 2004. In 2004, the three data sets dated 12 Jan. 2004, 26 Apr. 2004, and 31 May 2004, corresponding to winter (Jan.) and spring (Apr. and May), and their colour compositions, confirm the origins of changes to rain precipitation. This was due to the high signal strength of the 2004 winter season, which was translated into very high humidity compared to the two months of Apr. and May 2004. High intensity indicated by small signal dominance is observed, corresponding to that in the image acquired on 12 Jan. 2004, compared to the two other images acquired on 31 Jan. 2005 and 21 Jan. 2008. This may have been the result of high precipitation observed in this period for 2004 compared with 2005 and 2008 during the same month (Jan.). The changes in these diachronic images were observed in the lowest areas, such as those where wadis and agricultural fields occurred. The mountainous areas (high altitudes) have been preserved due to the low intensity of rainfall recorded during these study periods. Accordingly, it can be stated that this area was morphologically stable during the 12-year period. The irregular slope factor contributed to the transport of carved materials from the highest to the lowest elevation. On the slope surface, several valleys and tributaries (Wadis: Hanifa, Nisah, Al Awsat, Laha) are present, with beds carved out of solid and fragile formations, running eastwards, forming a dense hydrographic network. Various erosion factors, such as wind and water, have contributed to the morphological evolution of the region and the recharging of near-surface aquifers in highly permeable geological formations.

### 4 Discussion

Active remote sensing data are among the important sources for the study and analysis of natural hazards' occurrence. SAR is the effective, fast, and inexpensive all-weather operation data of active sensors on board orbital satellites and has the ability to be enhanced by several methods. It provides two components of the backscattered signal: energy (radiometry), which is transmitted and receives good signal amplitude, and time (phase), which relies on the recorded wave distance sent by the radar sensor and the one coming from the target. Amplitude characterises the high-frequency wave images which were used in several landforms and different applications. It can be used directly as a substitute for the optical image, reflecting the surface conditions. Stable wet surfaces are represented by high radiometry values; soft arid surfaces appear with low radiometry values. Therefore, radiometric corrections are very important before image visualisation and interpretation. ERS and ENVISAT are two important types of SER data, which provide valuable information with geological maps (including major geological structures such as faults, collapses, and valleys) and detect surface topography. The availability of ERS-1/2 and ENVISAT data at different dates contributes to risk management, both during and after disasters. Moreover, the diachronic analysis of these data highlights surface variations, particularly in river systems, depressions, fields, and urban areas.

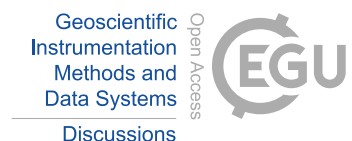

Radar images can detect mass movement, such as landslides and mudflows, by combining them with DEMs, and can suggest the direction of mass movement in the region accurately. The association of the amplitude image with a DEM has shown effectiveness in determining soil surface moisture in arid and semi-arid areas (Troufleau et al., 1994). Radar images have shown their usefulness in monitoring deforestation, ice melt, and the polar environment (Budkewitsch et al., 1996a, 1996b). They can

be used in studying and detecting mesoscale phenomena in oceans (Laborde and Deveaux, 1996), shape recognition and land-use planning (Rudant et al., 1996), and crop classification (Bruniquel and Lopes, 1994; Lopes and Sery, 1997). They have their own applications in geological and structural mapping (Singhroy and Saint-Jean, 1999; Wade, et al., 2001) and detection and characterisation of hydrocarbon slicks in the seas and oceans (Mercier et al., 2004). They are very strong tools in monitoring flood extent at the time of acquisition (Brivio et al., 2002; Mcmillan et al., 2006). Moreover, they are helpful in mapping urban

damage due to natural or industrial disasters and changes in coastal features (Ba et al., 2007). Synthetic-aperture radar (SAR) wavelengths and polarisations provide the best results for detecting, identifying, and mapping exposed and buried channels under the sand by detecting reflectance contrast differences between the bed of buried valleys or rivers and the surrounding environment. Utility, digital processing, visualisation, and interpretation of radar data (ENVISAT, ERS 1-2) at different dates and the application of differential interference methods, such as interferometric SAR (InSAR), DInSAR, small baseline subset

synthetic-aperture radar interferometry (SBAS-InSAR), and pulse-doppler radar (PS), were applied in several countries. Hachemi et al. (2009; 2010), Hachemi and Thomas, (2013), and Hachemi et al. (2014a, b; 2015; 2020) applied radar data in several countries such as France, Belgium, Romania, Canada, Romania, Vietnam, Mauritania, Morocco and Algeria. Diachronic DInSAR images draped over SRTM images are the most suitable data to detect and monitor risk areas, to map them as accurately as possible, and to estimate, if necessary, the resulting damage and landscape changes. Consequently, the SAR interferometry

technique has many applications in the area of deformation and geohazard monitoring, such as earthquakes, volacanoes, subsidence, and landslides (Graham, 1974; Zebker and Goldstein, 1986; Goldstein et al., 1988; Massonnet et al., 1993; Fruneau and Achache, 1995; Avallone et al., 1999; Briole et al., 1999; Fruneau and Sarti, 2000; Cakir et al., 2003; Closson et al., 2003; Stramondo et al., 2005; Delacourt et al., 2007; Beauducel et al., 2000; Hachemi, 2009; Hachemi et al., 2012). DInSAR is a means of detecting small displacements at the centimetre scale (1/2 wavelength applied in this study).

This study tests the ability of the SAR and DInSAR imageries and applies them on this arid region for the reported karstification (e.g. Bamousa et al., 2014) and recent activities of the Sahba fault and valley, to the east of Al-Kharj (Weigermars, 1998). The Sahba fault cuts through the biggest inland oil field (known as the Gawar oil field) and the Quaternary sand dunes, reaching the Arabian Gulf and connecting with the Zagros Mountain thrust belt. Also, with other regional Wadi Al Batin and Az-Zulfi faults forming a regional feature within Arabia, the East Arabian Block holds most of the oil and gas fields in the world

(e.g. Weigermars, 1998; Bamousa et al., 2017). Therefore, this study is conducted based on three factors. Firstly, there is the importance for the people of the oasis, built thousands of years ago for its mild weather, good resources of water, and agriculture, which are essential elements for living humans and animals. Secondly, the reported karstification, fault activities and the extensional geomorphology might have the potential for hazards, especially if triggered by artificial activities and expansion of



the nearby cities. Finally, the study area for the surrounding oil and gas fields has its place in the economy. Moreover, this study
also tests possible earthquake activities that might trigger reactivation of the faults and grabens by oil and groundwater extraction.

**5 Conclusion**

This study is a regional and remote sensing investigation of the morphology of an arid region, and it was possible to
clearly distinguish the different aspects that characterise the area. Multi-temporal analysis, conducted using amplitude images,
has shown the possibility of mapping surface changes at different dates. It also shows the ability to determine and locate faults,
fractures, and other geological features. It makes it possible to understand the evolution of drainage networks over different
periods. These radar images address the major issue of urban sprawl, which must be reviewed and planned in a preventive
manner. This research can serve as a tool for city planners and decision-makers to prepare future projects for the benefit of the
environment and society. Finally, this study contributes to the development of methodologies in similar conditions, including
spatial and temporal identification of areas potentially exposed to natural hazards. This study recommends preserving the region
of Wadi Nisah as a national geopark by the Ministry of Environment, Water and Agriculture for several reasons. In this way, all
municipalities would take this area out of plans for making new residential areas to expand the two cities of Riyadh and Al-
Kharj. The study area has the potential of groundwater, oil, and gas resources. The second potential is the faults' and grabens'
reactivations. The third potential is that, as an oasis that has been built thousands of years ago, it can be preserved for long-
standing food supply and therefore food safety. This study also recommends sustaining the groundwater for water safety by
changing the type and style of irrigated farms to another style that consumes less groundwater.

**Author contribution**

The authors conceived the idea presented and contributed to the formulation of the objectives and methodology of the research,
in particular the writing of the initial project. The first author was involved in the administration and supervision of the project
as well as in the funding acquisition and the implementation of the research schedule, leading to this publication. The second
author was responsible for the acquisition of the radar images and the numerical processing of the data and statistical calculations
(data curation). The third author suggested the study area for this project, and was involved in documenting the introductory parts
of the manuscript for his local background. All the authors contributed to the designing and describing of the carried-out
methodology, the investigation processes, formal analysis; they were involved in the verification, validation and interpretation
of the results and contributed to the writing of the final manuscript.

**Acknowledgments**

The research team would like to thank the Deanship of Scientific Research at King Abdulaziz University for its financial support
for this investigation (Project Number G-589-125-38). We also thank the European Space Agency (ESA) for providing us with
SAR images of the ERS and ENVISAT satellites as part of a research project (No. 28248). The work is based on the exchange
of experience between three universities (the Department of Geography and Geographic Information Systems, King Abdulaziz



University, Geology Department, Taibah University, and the Physical Geography Laboratory (LGP), CNRS, University
Paris1Pantheon-Sorbonne and University of Paris-Est in the field of remote sensing and field applications.

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




**Figure 1**: False-colour (2, 3, 4) Thematic Mapper Landsat imagery showing the study area, and main landscapes: Tuwaiq Mountain (TM), Wadi Nisah (WN), Al Khaj Depression (AD), Wadi Hanifah (WH) and Wadi Silay (WS).







**Figure 2:** Regional geology map of the study area (adapted from Bamousa et al., 2017).




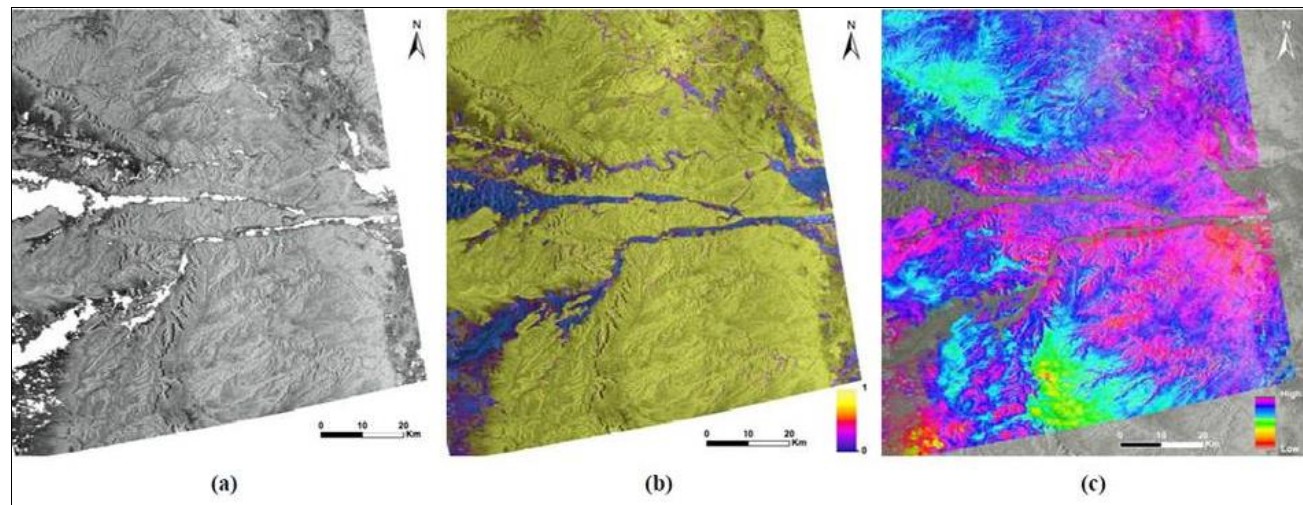

**Figure 3:** Differential interferometry SAR results of the 26/05/2004 and 31/01/2005 pair from the southern region of Riyadh; (a) amplitude; (b) coherence; (c) unwrapped phase difference (resolution 20 m).

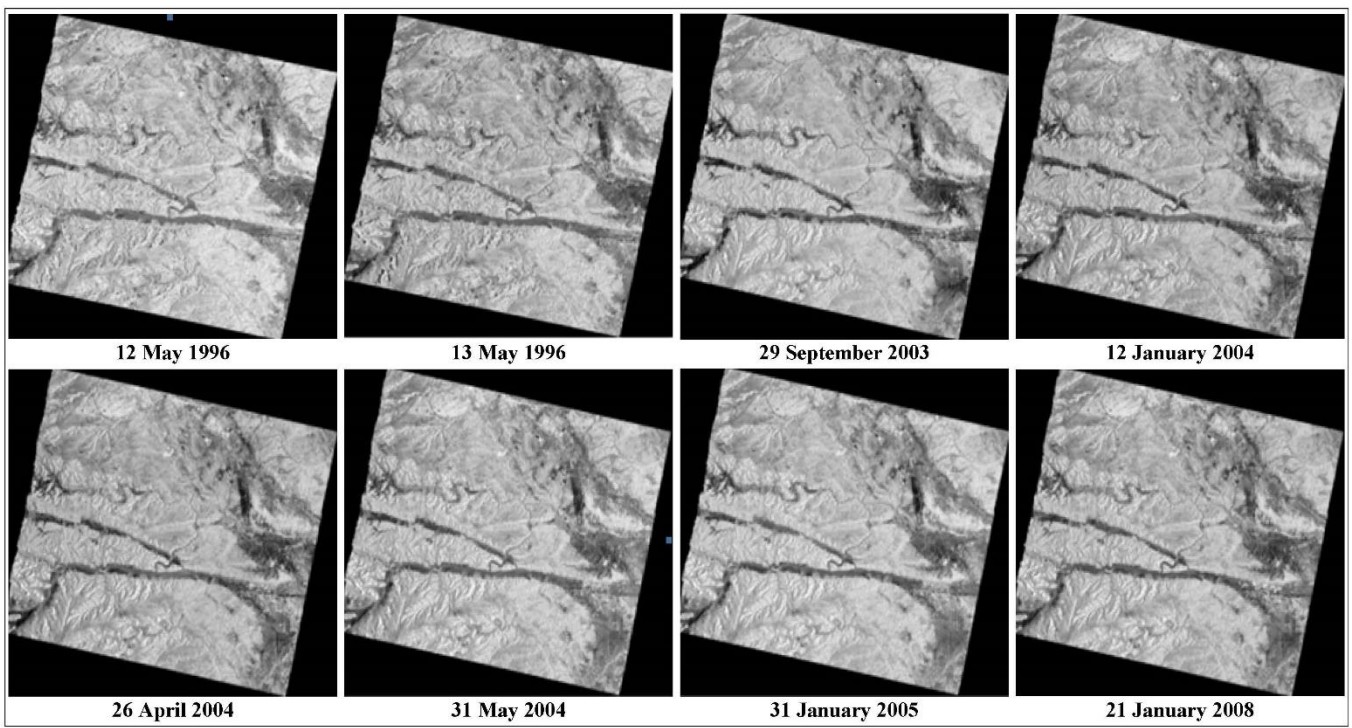


**Figure 4:** Intensity images of the study area at a resolution of 20 m, calibrated, georeferenced, orthorectified and filtered.

**Figure 5**: The colour compositions of calibrated, georeferenced and orthorectified intensity images of the study area at a resolution of 20 m (1996-2008).







**Figure 6:** False-colour (234) ETM+ Landsat imagery for 3 Apr. 2006 showing the development of the new system of hydrology towards Al Kharj and the sand dunes areas in the valleys and depressions.





**Figure 7**: Image intensity 13 May 1996 to 21 Jan. 2008 showing landscape morphology classification of the study area.


**Figure 8**: DEM topographic profiles of the intensity image; locations of these profiles are outlined in Fig. 7. Profile 1: across the Tuwaiq Mountain, cutting Wadi Laha, Nisah and Awsat grabens, from north to south. Profile 2 across the Tuwaiq Mountain, cutting Wadi Laha, Nisah and Awsat grabens, from north to south. Profile 3: across the Dughum depression and Nisah Graben. Profile 4: along Al Awsat Graben from west to east. Profile 5: along Nisah Graben from west to east.






**Figure 9:** The coherence results of the 26/05/2004 and 31/01/2005 pairs for the southern region of Riyadh (resolution 20 m).





**Figure 10:** The DInSAR interferometry results of the 26/05/2004 and 31/01/2005 pairs for the southern region of Riyadh (resolution 20 m). Altitude values ($x$) are not provided with a numbered scale for unknown coefficient ($k$) value in the following equation: $x = 2k\pi$. If $k$ for example is equal to 1, the lowest altitude value is equal to $2\pi$. Therefore, the interval of the provided range is equal to 50 m/colour, ranging from 400 m to 1,000 m according to the topographic map.





**Table 1:** Data used ERS-1/2 and ENVISAT.

| N° | Date | Time of day | Satellite | Orbit | Track | Frame | Center (Lat/Long) | Incidence angle |
|---|---|---|---|---|---|---|---|---|
| 1 | 12/05/1996 | 07:27:37,25 | ERS-1 | 25230 | 192 | 3123 | 24,36/46,56 | 23,247° |
| 2 | 13/05/1996 | 07:27:38,92 | ERS-2 | 05557 | 192 | 3123 | 24,22/46,53 | 23,247° |
| 3 | 29/09/2003 | 06:57:28,00 | ENVISAT | 08262 | 192 | 3118 | 24,28/46,60 | 22,931° |
| 4 | 12/01/2004 | 06:57:27,00 | ENVISAT | 09765 | 192 | 3123 | 24,26/46,59 | 22,939° |
| 5 | 26/04/2004 | 06:57:26,00 | ENVISAT | 11268 | 192 | 3115 | 24,25/46,59 | 22,942° |
| 6 | 31/05/2004 | 06:57:31,00 | ENVISAT | 11769 | 192 | 3123 | 24,25/46,59 | 22,929° |
| 7 | 31/01/2005 | 06:57:26,00 | ENVISAT | 15276 | 192 | 3120 | 24,25/46,59 | 22,922° |
| 8 | 21/01/2008 | 06:57:18,00 | ENVISAT | 30807 | 192 | 3119 | 24,25/46,59 | 22,921° |


**Table 2:** Geometric characteristics of each image tandem ERS-1/2 and ENVISAT. $B_{temp}$: Baseline time

$B_{perp}$: Perpendicular baseline Mod_Coh : Modeled coherence. $H_{am}$: Height of ambiguity [h = $H_{amb}$*phase/2π]. Delta fDC : Doppler baseline.

| N° | Dates (Master/Slave) | $B_{temp}$ (days) | $B_{perp}$(m) | Mod_Coh | $H_{amb}$(m) | Delta fDC (Hz) |
|---|---|---|---|---|---|---|
| | **ERS1/2** | | | | | |
| 1 | 12 May 1996/13 May 1996 | -1 | -169.96 | 0.68 | 55.66 | 278.56 |
| | **ENVISAT** | | | | | |
| 2 | 26 Apr. 2004/29 Sept. 2003 | 210 | -554.26 | 0.41 | 16.63 | 72.89 |
| 3 | 26 Apr. 2004/12 Jan. 2004 | 105 | 280.13 | 0.69 | -32.90 | -4.74 |
| 4 | 26 Apr. 2004/31 May. 2004 | -35 | 578.48 | 0.50 | -15.93 | 0.85 |
| 5 | 26 Apr. 2004/31 Jan. 2005 | -280 | -37.76 | 0.72 | 244.08 | 4.40 |
| 6 | 26 Apr. 2004/21 Jan. 2008 | -1365 | -161.99 | 0.01 | 56.89 | -27.41 |
| 7 | 12 Jan. 2004/29 Sept. 2003 | 105 | -830.25 | 0.26 | 11.08 | 77.64 |
| 8 | 12 Jan. 2004/31 May 2004 | -140 | 300.80 | 0.65 | -30.58 | 5.59 |
| 9 | 12 Jan. 2004/31 Jan. 2005 | -385 | -316.42 | 0.47 | 29.07 | 9.14 |
| 10 | 12 Jan. 2004/21 Jan. 2008 | -1470 | -439.45 | 0.01 | 20.93 | -22.67 |
| 11 | 31 Jan. 2005/29 Sept. 2003 | 490 | -516.71 | 0.30 | 17.83 | 68.50 |
| 12 | 31 Jan. 2005/31 May 2004 | 245 | 615.51 | 0.38 | -14.97 | -3.55 |
| 13 | 31 Jan. 2005/21 Jan. 2008 | -1085 | 124.37 | 0.01 | 74.09 | -31.81 |
| 14 | 29 Sept. 2003/31 May 2004 | -245 | 1130.84 | 0.04 | -8.16 | -72.05 |
| 15 | 29 Sept. 2003/21 Jan. 2008 | -1575 | 392.35 | 0.01 | -23.52 | -100.31 |
| 16 | 31 May 2004/21 Jan. 2008 | -1330 | -739.39 | 0.00 | 12.43 | -28.26 |



**Table 3:** Extreme cases of colours resulting from the coloured composition.

| a/b | 0 | 0 < β < 1 | 1 |
|---|---|---|---|
| 0 | Black | | Red |
| 0 < α < 1 | Intermediate colours between light blue and red | | |
| 1 | Light blue | | Yellow/white |


**Table 4:** The following legend shows the key to interpreting colour composition images in Figs. 5 and 7.

| | |
|---|---|
| 🟥 | Weak signal in 1st date and strong in 2nd date |
| 🟦 | High signal in 1st date and low signal in 2nd date |
| 🟨 | Strong signal in both dates (no change) |
| ⬛ | Weak signal in both dates (no change) |

**Table 5:** Rate of the extension of the city of Riyadh.

| Extension direction | UTM Coordinate Centre | | Length | Rate |
|---|---|---|---|---|
| | X (m) | Y (m) | (km) | (km/an) |
| South-East | 695670,87 | 2712995,17 | 2,83 | 0,24 |
| South | 679605,44 | 2712045,97 | 5,48 | 0,46 |
| South-West | 664306,42 | 2709481,76 | 9,23 | 0,77 |
| West-South-West | 652292,83 | 2716305,38 | 1,09 | 0,09 |