# Peer review of "Morphology of an arid landscape utilising synthetic-aperture radar (SAR) and differential interferometric SAR (DInSAR), southern Riyadh, Central Arabia"

_Geoscientific Instrumentation, Methods and Data Systems, 2020_

## Referee Comment (RC1) · Anonymous Referee #1 · 24 Oct 2020

General comments (G)

Although I have really appreciated the authors' efforts in applying complex techniques in the field of active remote sensing, I have noticed the following general issues in this manuscript:

General comment 1. Introduction. A big part of the introduction is only dedicated to the description of the study area. In scientific papers, I would expect: (I) general problem introduction; (II) at least a synthetic state of the art; (III) statement of the objectives;

(IV) description of the novelties introduced by this paper.

General comment 2. Data and method section should be improved with a better description of the implemented techniques and procedures; General comment 3. The discussion section does not provide links between results and objectives.

Specific comments:

Specific comment 1. lines 29-60 can be moved to a dedicated section (e.g. "Study area");

Specific comment 2. lines 80 – 81 " This makes it possible to measure vertical ground displacements up to 28 mm in the case of the C-band". Each fringe should be multiplied by 28 mm;

Specific comment 3. lines 74 -76 "It is directly usable and comparable to the optical image, and is also directly related to the surface conditions, thus, some surfaces are represented by a high radiometric value, such as water surfaces, whereas dry surfaces appear with low radiometric values". Normally, water surfaces have a low back-scattering.

Specific comment 4. Lines 95-100. Could you explicitly state the novelties introduced by this study?

Specific comment 5. Data and method section (see general comments). Moreover, I would suggest adding more details on the software (GAMMA right?) used (including a reference)

Specific comment 6. Figure 3. Scale bars are barely visible. Coordinates grid should be present;

Specific comment 7. Lines 139-145. Could you improve this paragraph considering: (I) an exhaustive description of equations; (II) a definition of the color of the pixel (RGB?);

Specific comment 8. lines 149-150. In the method section, the methodology for landscape detection is not introduced.

Specific comment 9. Figure 8. Abscissas and Ordinates are barely visible

Specific comment 10. line 170 "estimated". How?

Specific comment 11. line 175 "subtracted". Could you be more precise?

Specific comment 12. Lines 254-259. This is not a discussion related to your results.
* * *

---

## Referee Comment (RC2) · Anonymous Referee #2 · 21 Jan 2021

With reference to a Central Arabia landscape, the submitted manuscript would seek to achieve two goals, that are the SAR "ability" to detect changes in "morphology" and to assess some geohazards related to fault-activity and karst-dissolution. However, it is not well written and is rather confusing. It is not even clear the state of knowledge from which the authors started in their research and even what their actual findings are. Among the various primary and secondary aims that are stated in the course of the narrative, it would seem that perhaps that the one actually most relevant is to test "the ability of recent techniques and methodology to create short-period monitoring and

analyses via available SAR and DInSAR images" (lines 93-94; see also lines 260-262).

The poor structure of the manuscript is immediately evident from the first sentences of "1 Introduction", where instead of explaining the problem of general interest addressed, the authors first describe the study area. Please editing it. Furthermore, in 1 Introduction, the state of knowledge of the problems faced must be clearly explained.

If Fig. 3 "shows the primary results obtained from the pair of images obtained on 26 May 2004 and 31 Jan. 2005" (lines 124-125), why is it mentioned in section 2 (Data and methods)?

The whole subsection 3.1 does not seem a result of the use of SAR methods, rather a part of the state of knowledge. They must be rewritten.

I find that even in subsection 3.2 there is little clarity. It is not clear what are the previous knowledge and what are the findings of the research conducted by the authors. Among the various ambiguous sentences I point out the following: "The maximum height of elevation is approximately 1,000 m in the west, and the lowest altitude is approximately 400 m in the east, indicating a maximum 600 m of downthrown displacement and depression. Profiles 4 and 5, plotted E-W along the Awsat and Nisah valleys reveal eastward tilting and consequent capturing of the hydrologic system by the last tectonic event in the Central Arabian graben system (Fig. 8)" (lines 177-180). Do the altitudes and differences in height result from the "Amplitude and intensity images"? Were they not known before the authors' study? Again, are the "reveal eastward tilting and consequent capturing of the hydrologic system" a result of the submitted manuscript? Or are they the result of published studies?

In section 3.3 some repetitions are apparent. I am referring to good-poor coherences and their interpretations (lines 185-192). Please make it more understandable.

There must be consequentiality among state of knowledge and related issues to be faced, methodology, results, and everything must meet in the final discussion. This,

unluckily, does not appear in the submitted manuscript. Much of section 4 Discussion is a review of the uses of SAR methods, with no connection to the investigations carried out (lines 224-259). After, the authors stated again that their "study tests the ability of the SAR and DInSAR imageries and applies them on this arid region for the reported karstification (e.g. Bamousa et al., 2014) and recent activities of the Sahba fault and valley ..." (lines 260-265). Please, where are the results of these tests shown? Even the three factors stated in the final part of 4 Discussion (lines 260-270) are not related to what is stated in 3 Results. For example, where are the earthquake-activities tests described? Does the "reactivation of the fault" result from the use made of "short-period monitoring and analyses via available SAR and DInSAR images"?

The manuscript must be profoundly amended to be eligible for publication.

---

## Author Comment (AC1) · 15 Feb 2021

Dear Associate Editor

First of all, we would like to thank the first reviewer for considering revision, and for his/her valueble comments. Also we are grateful to his appreciation of our effort in applying a compolex techniques in the field of active remote sensing. Regarding general comment 1, we think that the introduction can be separated as he suggested into two sections to put the study area section immediatly as after introduction. General

comment 2, We have added more information about the software. General comment 3, The discussion has been improved to show linkage between results and objectives.

Specific comment 1, is applicable and was changed in the revised manuscript Specific comment 2, we have checked the statement retyped a revised one. Specific comment 3, also the statement was revised Specific comment 4, is also been restated to show the contribution of this study. Specific comment 5, more details on the software GAMMA is added with reference Specific comment 6 and 9, we put the scale and coordinates of the images in the figure, and removed the minimized image and a larger image. Specific comment 7, the equastion was revised Specific comment 8, the intrductory statement was revised for taking off the misunderstanding, occured in that section. Specific comment 10, we ll mentionhow it was estimated: from city limits difference between 1996 and 2008. Specific comment 11, the world substracted has been changed Specific comments 12, the entire discussion section is revised to be linked with other parts of the manuscript. The references mentioned in this comment is removed.

All changes will be shown in red color.

Regards

---

## Author Comment (AC2) · 15 Feb 2021

Dear Associate Editor

First of all, we would like to thank the second reviewer for considering revision, and for his/her valuable comments that enhanced the manuscript after revision. We noticed smiliarty of his/her revision with the first reviewer regarding the structures of the manuscript, which will be reorganized and mandated as suggested. We also put the statement of knowledge (sebsection 3.1) in the section of study area, which was pro-

posed by the first reviewer. Now the entire section (study area) would represent statement of knowledge. We have also put what is this study is about, at the beginning of the manuscript as suggested by the second reviewer. Figure 3 was removed from its place and put in the results section. Regarding the results of the topographic profiles (Fig. 8 in the submitted manuscript), this results are new data but because other people mentioned this difference in altitude about 500 m, and we found it 600 m, however we mentioned the previous work studies. Regarding the capturing of the old drainage system, two references have mentioned this before, and will be shown in the revised manuscript, but this study shows different capturing direction, however, we mentioned the previous work studies. We have also clarified all misunderstanding statements, mentioned by the reviewer in the third section of the submitted manuscript. It will be the fourth one after adding a new section for study area. The reactivation statement has caused misunderstanding, therefore, it will be clarified and restated in the revised manuscript, at the end of section 4 (the discussion). The section of discussion has been totally revised to be in accordance with previous sections, also, many unrelated references were removed.

Regards,